# Vasoactive Intestinal Polypeptide (VIP) in the Intestinal Mucosal Nerve Fibers in Dogs with Inflammatory Bowel Disease

**DOI:** 10.3390/ani10101759

**Published:** 2020-09-28

**Authors:** Andrzej Rychlik, Sławomir Gonkowski, Jarosław Całka, Krystyna Makowska

**Affiliations:** 1Department of Clinical Diagnostics, Faculty of Veterinary Medicine, University of Warmia and Mazury in Olsztyn, Oczapowskiego 14, 10-957 Olsztyn, Poland; andrzej.rychlik@uwm.edu.pl; 2Department of Clinical Physiology, Faculty of Veterinary Medicine, University of Warmia and Mazury in Olsztyn, Oczapowskiego 13, 10-957 Olsztyn, Poland; slawomir.gonkowski@uwm.edu.pl (S.G.); jaroslaw.calka@uwm.edu.pl (J.C.)

**Keywords:** digestive tract, dogs, enteric nervous system, immunohistochemistry, VIP

## Abstract

**Simple Summary:**

Canine inflammatory bowel disease (IBD)—a group of gastrointestinal disorders—is a serious problem in veterinary medicine. The etiology of IBD remains unknown, and its diagnosis and effective treatment are difficult. One of the less-known aspects of IBD pathology is the influence of this disease on the enteric nervous system, which is located in the intestinal wall and regulates most of the gastrointestinal functions. Therefore, the aim of the present study was to evaluate the influence of IBD on the intramucosal nerve fibers containing vasoactive intestinal polypeptide (VIP). VIP is one of the most important substances produced by the enteric nervous structures that is involved in many regulatory processes in the gastrointestinal tract. The obtained results show that IBD induces changes in the density of intramucosal VIP-positive nerve fibers in the canine gastrointestinal tract. It suggests that VIP is involved in the pathological processes occurring during this disease. Observed changes may be a result of neuroprotective and/or adaptive processes regulated by VIP, aimed at the homeostasis maintenance in the inflamed gastrointestinal (GI) tract and induced by proinflammatory factors.

**Abstract:**

Canine inflammatory bowel disease (IBD) is a group of enteropathies with nonspecific chronic symptoms and poorly understood etiology. Many aspects connected with IBD are not understood. One of them is the participation of the intestinal nervous system in the development of pathological processes. Thus, this study aimed to demonstrate changes in the density of intramucosal nerve fibers containing vasoactive intestinal polypeptide (VIP)—one of the most important intestinal nervous factors caused by the various stages of IBD development. Mucosal biopsy specimens collected from the duodenum, jejunum and descending colon of healthy dogs and dogs with varied severity of IBD were included in the experiment. The density of VIP-like immunoreactive (VIP-LI) nerves was determined by a single immunofluorescence technique and a semi-quantitative method consisting in VIP-LI fiber counts in the field of view (0.1 mm^2^). The obtained results indicate that IBD induces changes in the density of mucosal VIP-LI nerve fibers in the canine gastrointestinal tract. The initial decrease is followed by an increase in VIP-like immunoreactivity in successive stages of the disease. These observations show that VIP is a neuronal factor that participates in the pathological processes connected with canine IBD. The observed changes probably result from the neuroprotective and/or adaptive properties of VIP. Protective and adaptive reactions induced by inflammation aim to protect the GI tract against damage by proinflammatory factors and ensure the homeostasis in the enteric nervous system (ENS) under the conditions changed by the disease process.

## 1. Introduction

Canine inflammatory bowel disease (IBD) is a group of idiopathic and inflammatory enteropathies characterized by persistent or recurring gastrointestinal symptoms. Since knowledge concerning IBD is rather scarce, this disease is a serious problem in modern veterinary. To date, the etiology of the disease remains unknown. According to the current knowledge, IBD results from a combination of genetic factors, environmental factors (such as gut microbiota and food antigens) and inadequate host responses to gastric contents and selected drugs [1,2,3,4]. The main clinical symptoms of IBD are vomiting, diarrhea, weight loss, bloating, abdominal cramping and loss of appetite [5]. Clinical symptoms are accompanied by histopathological changes in the small intestine and/or the colon. Recent studies have shown that substances acting as neurotransmitters and/or neuromodulators in the enteric nervous system (ENS) and extrinsic innervation of the gastrointestinal tract could be implicated in the pathogenesis of intestinal inflammations [6,7,8]. However, the exact roles of the majority of active neuronal substances in the pathophysiology of IBD have not been yet elucidated. 

One of the neuronal substances that are widely distributed in the intestinal innervations, and which may be involved in mechanisms connected with enteropathies, is vasoactive intestinal polypeptide (VIP) [9,10].

VIP was first isolated in 1970 by Said and Mutt [11]. This polypeptide is composed of 28 amino acids. In the ENS, VIP relaxes smooth muscles and regulates the secretory activity of digestive organs [9,10,11,12,13]. Its influence on the enteric mucosa differs in various segments of the gastrointestinal tract. For example, VIP inhibits the secretion of gastric acid in the stomach, but it also stimulates the release of intestinal juice and bile in the intestines [14]. VIP is also an immunomodulatory factor that exerts anti-inflammatory effects by inhibiting the release of proinflammatory cytokines and mediators such as interleukin (IL)-6, IL-12, tumor necrosis factor alpha (TNF-α), nitric oxide and chemokines [15,16,17,18]. The peptide acts on VPAC_1_ and VPAC_2_ receptors and promotes the differentiation of T-helper (Th) cells into type 2 T-helper (Th2) cells [17,18,19]. VIP also stimulates the production of regulatory T cells and inhibits macrophage proinflammatory activities [9]. Moreover, previous studies have shown that VIP could also be implicated in neuroprotective responses in the ENS [20]. The above hypothesis may be supported by the increase in VIP-like immunoreactivity within the ENS in patients suffering from Crohn’s disease or ulcerative colitis [21,22,23]. These changes in the number of VIP-positive enteric nervous structures during gastrointestinal diseases may suggest that the examination of VIP—like immunoreactivity—could have diagnostic value.

The objective of this study was to determine, for the first time, the changes in the number of VIP-LI nerve fibers in the mucosa of different sections of the canine gastrointestinal tract caused by IBD with various levels of severity. The results will broaden the knowledge of the role of VIP in the ENS in pathological processes occurring in canine IBD.

## 2. Materials and Methods

Twenty-eight German shepherd crossbreed dogs of both sexes were included in the present study. The dogs weighed from 15 to 25 kg and were aged from 6 to 10 years. The control group was formed from healthy dogs (n = 7) included in the study during the IBD screening tests conducted in a dog shelter in Olsztyn (Poland). Animals suffering from IBD within the experimental groups were private-owned patients of the Veterinary Clinic of the University of Warmia and Mazury in Olsztyn. All actions during the present experiment were conducted in accordance with the requirements of the Local Ethics Committee for Animal Experimentation in Olsztyn (Decision No. 47/2009/DTN). The qualification of animals to experimental groups was done based on the results of clinical, laboratory, endoscopic and histopathological examinations.

The dogs were subjected to a comprehensive diagnostic procedure, including clinical, laboratory, endoscopic and histopathological examinations of the intestinal mucosal layer. Systemic diseases, antibiotic response enteropathy (ARE), food response enteropathy (FRE) and parasitic infestations were excluded in all dogs according to the recommendations of the World Small Animal Veterinary Association Gastrointestinal Standardization Group. Seven healthy dogs, screened with the same tests as in the experimental group, were included in the control group.

Dogs suffering from IBD were divided into three groups (seven animals in each) based on their canine IBD activity index (CIBDAI) scores [24]. The groups were as follows: group I—dogs with mild disease process (CIBDAI score—4–5 points and histopathological score “+”), group II—moderate disease process (CIBDAI score—6–8 points and histopathological score “++”) and group III—severe disease process (CIBDAI score—10–16 points and histopathological score “+++”). Only dogs with inflammatory changes simultaneously present in the duodenum, jejunum and colon were included into the experiment.

Specimens for immunofluorescence analyses were collected from all dogs included in the experiment (both from the control and experimental groups) during gastroduodenoscopy and colonoscopy using FB-24U-1 biopsy forceps with a diameter of 2.5 mm and FB-50U-1 biopsy forceps with a diameter of 3.7 mm (Olympus). The collected fragments of the intestinal mucosa (three specimens from every intestinal part studied) were subjected to fixation in a 4% buffered paraformaldehyde solution for 15 min. After this time, the tissues were rinsed in a phosphate solution (pH 7.4) for three days at 4 °C. The specimens were then put into an 18% buffered sucrose solution and stored for three weeks at 4 °C. After three weeks, the fragments of the mucosal layer were frozen in −20 °C, cut in 10-μm sections using the Microm cryostat (HM525, Walldorf, Germany) and mounted on microscopic slides. The intestinal sections were subjected to a routine single immunofluorescence technique described previously by Rychlik et al. [8] with the use of commercial antibodies. In brief, this method consisted of the following stages: (1) drying for 45 minutes at room temperature; (2) incubation with a blocking solution containing 10% goat serum, 0.1% bovine serum albumin (BSA), 0.01% NaN_3_, Triton X-100 and thimerosal in phosphate-buffered saline (PBS) for 1 h; (3) the incubation overnight in a humidity chamber at room temperature with polyclonal anti-VIP antibody (rabbit, Cappel, Aurora, OH, USA, catalogue No. 11428, working dilution 1:5000); (4) incubation with a specific secondary antibody conjugated to Alexa Fluor 594 (donkey, anti-rabbit, Invitrogen, Waltham, MA, USA, working dilution 1:1000) for 1 h at room temperature to visualize the “antigen–primary antibody” complexes and (5) the coating of the specimens with a glycerol solution in PBS (1:2; pH 7.4) and covering them with coverslips. Between particular stages, the slides with specimens were rinsed in PBS (0.1 M, pH 7.4, 3 × 15 min.).

Stained sections were evaluated under an Olympus BX51 fluorescence microscope (Tokyo, Japan) equipped with appropriate filters. VIP-like immunoreactive (VIP-LI) nerve fibers localized in the mucosal layer were subjected to a semi-quantitative evaluation by counting their numbers in the field of microscopic view (0.1 mm^2^). The number of fibers was evaluated in four fields of view in three sections of every biopsy specimen from every intestinal segment included in the study (duodenum, jejunum and colon). In every dog in the experiment (both in the control and suffering from IBD), a total of 36 fields of view in every segment were evaluated. The microscopic fields included in the study were located at least 100 µm from each other to avoid repeated counts. The results were grouped and the mean values and standard deviation were calculated. The specificity of the used primary antibody was verified in a pre-absorption test. A synthetic peptide (cat no. V6130, Sigma, St Louis, MO, USA) was added in the amount of 10 µg, to 100 µl of the antibody solution with a working concentration. This mixture was incubated for 24 hours at 4 °C and used to stain sections of small and large intestines. The pre-absorption test completely eliminated specific staining. Moreover, typical replacement and omission tests were performed.

The significance of differences between groups was determined by the Kruskal-Wallis test at *p* ≤ 0.05 (significant) and *p* ≤ 0.01 (highly significant). The results were processed in the Statistica 9.1 application (StatSoft, Inc., Cracow, Poland).

## 3. Results

In dogs suffering from IBD included in the study, a close correlation between symptoms evaluated in the CIBDAI scale (such as patience activity, loss of appetite, frequency of vomiting, frequency of defecation and fecal consistency, as well as loss of body weight) and the severity of the disease was observed. This correlation was especially clear for the frequency of defecation and fecal consistency. Along with the severity of IBD, the frequency of defecation was higher, and feces were watery. Moreover, in animals with severe IBD, blood in the stool was sometimes observed.

Mucosal nerve fibers containing VIP were noted in the mucosal layer of all studied fragments of the intestines, both in physiological conditions and in animals suffering from IBD (Table 1 and Figure 1).

In healthy dogs, the average number of the mucosal nerve fibers immunoreactive to VIP in the field of view was similar in all fragments of the digestive tract studied and amounted to 25.12 ± 3.16, 25.85 ± 2.29 and 26.27 ± 1.9 in the duodenum, jejunum and colon, respectively (Figure 1C). In dogs with mild IBD (group I), the number of VIP-positive nerves in the evaluated segments of the intestinal tract was slightly lower compared to the control dogs (Figure 1I), but statistically significant differences between these groups were not observed (Table 1) The average number of intramucosal VIP-LI nerves per observation field in dogs with mild IBD amounted to 23.01 ± 2.10 in the duodenum, 24.49 ± 1.91 in the jejunum and 24.24 ± 2.47 in the descending colon (Table 1).

In turn, in the dogs suffering from moderate IBD (group II), a statistically nonsignificant increase in the average number of VIP-LI fibers was noted in the duodenum and jejunum in comparison to the control group and animals with mild IBD (Figure 1II). The average number of intramucosal VIP—positive nerve fibers in the above-mentioned intestinal segments achieved 27.47 ± 1.71 and 26.42 ± 2.00, respectively. The number of investigated fibers in the descending colon of animals in group II amounted to 27.20 ± 0.61, and this value was statistically significantly different from the values observed in the dogs of group I (Table 1). 

The largest number of VIP-LI intramucosal nerves was noted in dogs suffering from severe IBD (group III) (Table 1 and Figure 1III). In the duodenum, the average number of such fibers amounted to 30.10 ± 2.80 nerves per observation field, and this value was statistically significantly higher compared to values noted in the duodenum of control animals and dogs with mild IBD (group I). The average number of nerves immunoreactive to VIP in the jejunal mucosal layer in dogs of group III was slightly lower than the number of nerves observed in the duodenum and reached 28.44 ± 2.38. This value was statistically significantly higher compared to values noted in the animals of group I. In turn, the average number of VIP-LI fibers in the colons of dogs suffering from severe IBD amounted to 27.19 ± 5.19, and in this segment of the intestine, statistically significant changes between group III and the other groups of experimental animals were not observed (Table 1).

During the present experiment, differences in the morphological characteristics of VIP-positive nerves between particular groups of animals, as well as between the intestinal segments studied, were not observed. Both in animals under physiological conditions and dogs suffering from IBD of all severity grades, the intramucosal nerves immunoreactive to VIP in all segments of the intestine were thick and well-visible and showed varicosities and formed bundles (Figure 1).

During the histopathological examination of the duodenum (Figure 2D) and jejunum (Figure 2J) in dogs suffering from mild IBD, the following changes were observed: (1) villi were deformed (long and pointy in four dogs and short and coryneform in three dogs), (2) the structure of the epithelium was saved in five dogs, whereas it was changed in two dogs, (3) a slight decrease in the number of goblet cells with an increase in the number of endothelial lymphocytes, and minor lymphocytic and plasmocytic infiltration within the lamina propria in all animals of group I, was noted, (4) no numerous fibroblasts were noted in the lamina propria and (5) congestion and dilation of lymphatic vessels was noted. In the descending colon, there was an increased amount of fibrous connective tissue between glands of the mucosal layer and dilation of blood vessels, as well as lymphocytic and plasmocytic infiltration.

In dogs suffering from moderate IBD, histopathological changes were more visible (Figure 2). In the duodenum and jejunum, the villi were conical and congested. Moreover, a moderate number of endothelial lymphocytes, as well as a small number (smaller than in the control group and group I) of goblet cells, were noted. Numerous lymphatic and plasmocytic cells were present in the lamina propria, which were accompanied by the lymph stasis and dilation of lymphatic vessels. In the descending colon (Figure 2C), numerous exfoliated epithelial cells were observed. The colonic mucosal glands contained a large volume of mucus and were surrounded by thick strands of the fibrous connective tissue. Moreover, numerous lymphocytic and plasmocytic cells were noted between the colonic mucosal glands.

In the duodenum and jejunum of dogs with severe IBD, the intestinal villi were short (in two dogs) or thin and long (in five dogs). In all animals, the villi considered a large quantity of fibrous connective tissue and a large number of lymphocytes. The number of goblet cells was lower than in other groups of animals. Massive infiltration of the lymphocytic and plasmocytic cells in the lamina propria was also noted. In the descending colon, a large amount of mucus and exudate was observed. Colonic mucosal glands were surrounded by thick strands of fibrous connective tissue (the amount of this tissue was greater than in the animals of group II). Massive lymphocytic and plasmocytic infiltration in the lamina propria of the colonic mucosal layer was also observed.

## 4. Discussion

Based on previous studies, VIP is one of the most important neuromediators and/or neuromodulators in the nervous structures supplying the gastrointestinal tract. Until now, this polypeptide, built of 28 amino acid residues and affecting two types of G protein-coupled receptors (VPAC1 and VPAC2), has been described in neurons and nerves innervating the stomach and intestine in numerous mammals species, including humans [25,26]. In the digestive tract, VIP is known, first of all, as a potent inhibitory factor, which causes the relaxation of smooth muscles and, therefore, contributes to the halting of peristalsis and widening of blood vessels located in the mesentery and the wall of the stomach and intestine [12,13,14,15,16,17,18,19,20,21,22,23,24,25,26,27]. It is also known that VIP may be involved in the regulation of the secretory activity in the digestive tract and processes connected with immunological and/or inflammatory reactions [9]. Moreover, previous studies have reported that VIP participates in protective and adaptive processes leading to the maintenance of homeostasis during various pathological processes and ensuring the functions of the ENS in conditions changed by the disease process [10,11,12,13,14,15,16,17,18,19,20,21,22,23,24,25].

These mentioned-above multidirectional effects of VIP on the GI tract are possible thanks to the wide spread of VIP and its receptors in the stomach and intestines that have been found in the GI tract of numerous mammal species, including humans [25,26,27,28]. Similar to other species, VIP has also been described in the various segments of the canine GI tract, especially in the gastrointestinal sphincter muscles [29,30,31]. Previous studies have reported that VIP is present in various parts of the canine ENS [32,33,34,35]. It is known that about 20% of neurons located in the myenteric plexus and about 50% of cells in the submucous plexus show the presence of VIP [32]. Moreover, the presence of VIP has been described in the density network of nerve fibers supplying various parts of the intestinal wall, including the mucosal layer, blood vessels in the submucosa and circular and longitudinal muscular layers [29,30,31,32,33,34,35,36]. Previous studies have reported that VIP receptors are also widely distributed in the canine GI tract. Their localization depends on the segment of the GI tract, but generally, they are present in the mucosal layer, the wall of the submucosal blood vessels and synaptosomes and intestinal lymph nodules, as well as in the muscular layer [30,31,32,33,34,35,36,37,38].

The present study confirms that VIP is present in a relatively high number of the enteric nervous structures in dogs and may participate not only in the regulation of the GI tract functions under physiological conditions but, also, play some roles during inflammatory processes. Interestingly, although the previous studies have described the presence of VIP in the intestinal endocrine cells, as well in the inflammatory cells [28,29,30,31,32,33,34,35,36,37,38,39,40], during the present investigation, VIP both in the control animals and in dogs with IBD were found only in the intramucosal nerve fibers.

Changes in the number of VIP-LI intramucosal nerves observed during IBD in dogs may have resulted from both the participation of VIP in inflammatory processes and the involvement of this substance in protective reactions. This is all the more likely since, in the light of the previous investigations, VIP is a relatively potent factor influencing the macrophage activity, causing a decrease in proinflammatory factor production [17,18,19,20,21,22,23,24,25,26,27,28,29,30,31,32,33,34,35,36,37,38,39,40,41]. It is also known that this peptide plays an important role in neuroprotective processes by the impact on glial cells and their stimulation to neurotrophic factor synthesis [42]. These observations, together with the results obtained during the present study, showed an increase of the number of VIP-positive nerves in animals. These inflammatory histopathological changes (such as lymphatic and plasmocytic infiltration and numerous exfoliated epithelial cells) strongly suggest that changes in the number of VIP-positive intestinal nervous structures are connected with the inflammatory processes and/or protective functions of VIP. Moreover, the correlation of a high number of VIP-positive nerves with the hyperemia of the intestinal mucosal layer may confirm the participation of VIP in vasodilation known from previous studies. The next reason for the observed changes may be the participation of VIP in the intestinal secretory activity, because the higher number of VIP-positive nerves was noted in dogs suffering from severe IBD in which, during a histopathological examination, an increase in the production of mucus was noted, and one of the symptoms was watery feces. However, it seems that the participation of VIP in intestinal motility plays a reduced role in the changes noted during the present study, because it is known that nerve fibers located in the mucosal layer mainly regulate the mucosal layer activity and protective processes and are not involved in the processes connected with muscular activity. Nevertheless, the exact reason for the changes noted in the present investigation remain unclear and may be connected with various mechanisms, such as an increase in protein synthesis at the stage of transcription and/or translation in response to inflammatory processes, as well as inflammatory-induced disturbances in the intracellular transport of VIP from neuronal cell bodies to the nerve endings.

It should be noted that previous studies have reported changes in the expression of VIP in the digestive tract during various intestinal diseases. Significant changes in the VIP expression have been observed both during “naturally” occurring pathological states and in experimental processes. For example, such changes have been noted in patients with Crohn’s disease [23] and people suffering from ulcerative colitis [43,44]. Moreover, significant correlations between the VIP levels in the serum and the severity of intestinal inflammation have been also observed [45], which suggests that the testing of VIP levels may be one of the diagnostic factors during gastrointestinal diseases. Visible changes in the number of VIP-LI intestinal neuronal structures was also noted during experimental inflammation and after damage to the nerves supplying the gastrointestinal tract [25]. Interestingly, in light of the previous studies, changes in the VIP–like immunoreactivity in the innervation of the gastrointestinal tract depends on the characteristics of the disease. For example, in people suffering from Crohn’s disease, an increase in the population of VIP-LI nervous structures has been noted [22], while ulcerative colitis in children caused a decrease in the number of intramucosal VIP-LI nerve fibers in the colon [46]. To date, the reasons for these differences have not been elucidated.

Since VIP shows neuroprotective properties [15,16,17,18,19,20], controls the immune reactions [41] and plays a key role during inflammatory processes [25,26,27,28,29,30,31,32,33,34,35,36,37,38,39,40,41,42,43,44,45,46,47,48], the possibility of the use of VIP analogs as drugs during intestinal inflammations has been tested in recent years. The results showed that VIP analogs ameliorated changes in the colon after experimentally induced colitis in rats and reduced necrosis, hyperemia and swelling [24].

Until now, the possibility to use VIP in the treatment of IBD is suggested by some investigations on experimentally induced intestinal diseases in rodents. Namely, it is known that the administration of VIP (especially using sterically stabilized micelles) is capable of alleviating colitis in mice and showing anti-inflammatory and antidiarrheal effects [49]. Therapeutic effects of VIP may result from VIP-induced stabilization of the intestinal immune homeostasis [50], protection of the epithelial mitochondria [51], roles in the maintenance of an appropriate intestinal barrier integrity [52,53] and blockage of the inflammatory cascade [54]. However, it should be underlined that the therapeutic roles of VIP in experimentally induced colitis in rodents are controversial, because other studies (contrary to the above-mentioned) strongly suggest proinflammatory and pathogenic roles of VIP and suggest the use of VIP antagonists and VIP receptor blockades in the treatment of intestinal inflammatory processes [55]. Therefore, the eventually application of VIP in the treatment of canine IBD requires further intensive clinical research.

## 5. Conclusions

The results obtained during the present study showed that canine IBD may affect the number of VIP-LI nerve fibers in the intestinal mucosal layer, especially when the disease process is severe. Most likely, the observed changes result from adaptive, neuroprotective and/or anti-inflammatory actions of this peptide, but the exact mechanisms influencing the number of VIP-positive nerves remain unclear. Nevertheless, observations made during the present experiment strongly suggest the participation of VIP in mechanisms connected with canine IBD and, therefore, may be the first step (after further studies) toward the introduction of VIP analogs in the treatment of this disease in the future. On the other hand, since the changes noted in the number of intramucosal VIP-positive nerves are relatively insignificant and mainly concern the severe form of this disease, the examination of VIP levels in the serum and/or the number of VIP-LI structures within the intestinal specimens during diagnosis of this disease is a matter of debate today and requires further comprehensive studies.

## Figures and Tables

**Figure 1 animals-10-01759-f001:**
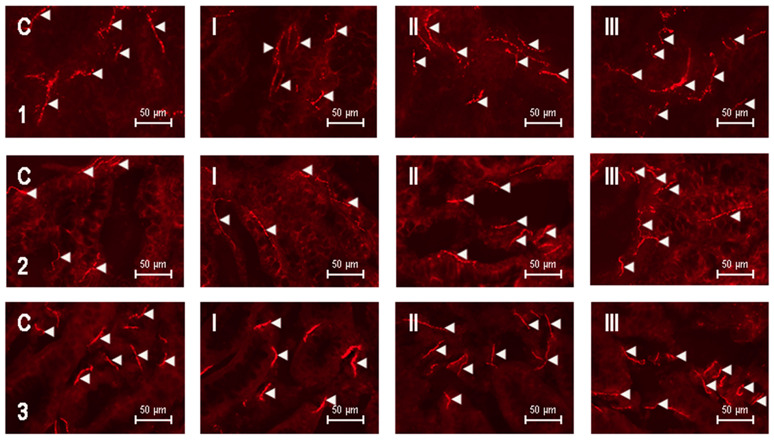
Vasoactive intestinal polypeptide (VIP)—positive nerves (indicated with arrowheads) in the mucosal layer of the canine digestive tract: **1**—duodenum, **2**—jejunum, **3**—descending colon, **C**—control dogs, **I**—dogs with mild inflammatory bowel disease (IBD), **II**—dogs with moderate IBD and **III**—dogs with severe IBD.

**Figure 2 animals-10-01759-f002:**
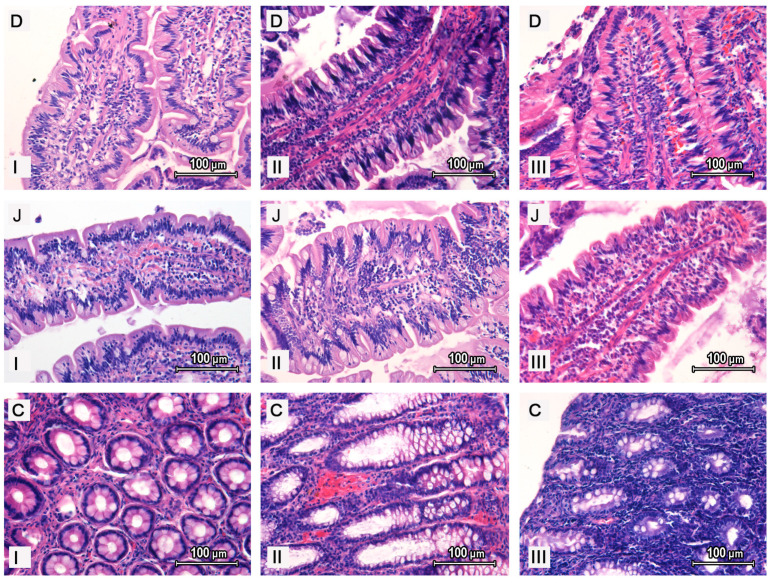
Histopathological examination of the duodenum (**D**), jejunum (**J**) and descending colon (**C**) of dogs with mild (**I**), moderate (**II**) and severe (**III**) IBD.

**Table 1 animals-10-01759-t001:** Inflammatory bowel disease (IBD)-induced changes in the number of intramucosal vasoactive intestinal polypeptide (VIP)-like immunoreactive nerve fibers in one field of view in the duodenum, jejunum and descending colon of the control group (C) dogs and dogs with mild (Group I), moderate (Group II) and severe (Group III) IBD.

Intestinal Segment	Group C	Group I	Group II	Group III
Duodenum	25.12 ± 3.16 ^D^	23.01 ± 2.10 ^D^	27.47 ± 1.71	30.10 ± 2.80 ^AB^
Jejunum	25.85 ± 2.29	24.49 ± 1.91 ^D^	26.42 ± 2.00	28.44 ± 2.38 ^B^
Colon	26.27 ± 1.9	24.24 ± 2.47 ^C^	27.20 ± 0.61 ^B^	27.19 ± 5.19

Values are shown as the average number of fibers ± SEM per field of view in the control group (C) and dogs with mild (Group I), moderate (Group II) and severe (Group III) IBD, A—significantly different from control, B—significantly different from group I, C—significantly different from group II and D—significantly different from group III. Kruskal-Wallis test: *p* < 0.05—lowercase letters and *p* < 0.01—uppercase letters.

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
