# Peer review of "Vasoactive Intestinal Polypeptide (VIP) in the Intestinal Mucosal Nerve Fibers in Dogs with Inflammatory Bowel Disease"

_animals, 2020, doi:10.3390/ani10101759_

Round 1

Reviewer 1 Report

Issues have been addressed and now seems acceptable for publication

Reviewer 2 Report

The authors accepted most of the suggestions and made changes to the manuscript, explaining better the flawed points mentioned in the review od the first version. This has improved the presentation and comprehension of the text and for this reason we recommend publishing it.

This manuscript is a resubmission of an earlier submission. The following is a list of the peer review reports and author responses from that submission.

Round 1

Reviewer 1 Report

The manuscript by Rychlik et al. describes interesting findings of potential involvement of VIP, in the pathogenesis of canine IBD. Although, there is merit in the work, improvement in data presentation and interpretation is required. Specific concerns are listed below;

  1. The manuscript lacks data. There is only one table and one figure. The authors should supplement the data with appropriate histological evaluations in each region (duodenum, jejunum and colon) and histopathological scores to supplement the current findings.
  2. The dogs which were used for the study diagnosed with IBD, it is unclear in which region the inflammation was present; thus, could the differences not observed for VIP immuno-reactivity in regions be related to that? please discuss and include which region of the intestine was affected with inflammation.
  3. The authors should be consistent when referring to the study subjects; they are referred to as "patients" (Page 9, Line161) and "dogs" (Page 9, line 165) in the manuscript. This is very confusing to the reader.
  4. In Figure 1 the staining is not very clear. Please include higher quality images and define the legends in a more informative manner.
  5. There are several relevant studies pertaining to identifying receptors in the intestine for VIP and abundance of VIP in canine intestine, these should be cited and discussed in the manuscript. 
  6. VIP is studied as a therapeutic for managing IBD in rodent models and have also shown deleterious effects. Please discuss this and relate it to the current findings in canine IBD.
  7. There are many grammar errors (Simple summary needs extensive revisions)and incomplete sentences (page 5 line 169) in the manuscript. Significant corrections are required.
  8. Please use histological representative images when describing data in page 5, it will add clarity to the findings.

Reviewer 2 Report

The aim of the present study was to evaluate how IBD affects nerve fibers containing vasoactive intestinal polypeptide (VIP) located in the mucous layer of the intestine in cases of canine inflammatory bowel disease. Idiopathic and inflammatory enteropathies are chronic and debilitating diseases that affect dogs and also humans. Studies like this are important to elucidate the pathogenesis and suggest treatment alternatives.

The subject has been the object of study by the group of researchers, which strengthens the consistency and accuracy of the methodology used, and there are already publications on other substances (chemical mediators, neurotransmitters) that may be involved in the pathogenesis of IBD. Therefore, it is basic research that can be applied in the development of therapies, as suggested by the authors themselves.

Lines 23-23 and 38-39 - Observed changes may be a result of neuroprotective and/or adaptive properties of VIP. This sentence has a vague meaning, which could be modified to better explain the neuroprotective/adaptive action as was done in the text (lines 237-240). “These inflammatory histopathological changes (such as lymphatic and plasmocytic infiltration and numerous exfoliated epithelial cells) strongly suggest that changes in the number of VIP-positive intestinal nervous structures are connected with the inflammatory processes and / or protective functions of VIP”.

What the authors suggest as an” adaptive action” is not clear to the reader.

Lines 217-228 - The text deals with a literature review on parameters that were not evaluated in this study. Then, on line 229 where the authors report that “The results described in this study confirm the parameters described above (referring to lines 217-228).

The description would be more objective if started with the text of lines 229-283, which follows as a very clear and interesting discussion about the results obtained.

As there are reports that inflammatory cells can also produce VIP, the authors would comment in the text whether or not there was also the marking of inflammatory cells in addition to nerve fibers.